# Intertypic Recombination Between Coxsackievirus A16 and Enterovirus A71 Structural and Non-Structural Genes Modulates Virulence and Protection Efficacy

**DOI:** 10.3390/vaccines13101017

**Published:** 2025-09-29

**Authors:** Hooi Yee Chang, Han Kang Tee, Kien Chai Ong, Kartini Jasni, Syahril Abdullah, I.-Ching Sam, Yoke Fun Chan

**Affiliations:** 1Department of Medical Microbiology, Faculty of Medicine, Universiti Malaya, Kuala Lumpur 50603, Malaysia; hooiyee@um.edu.my (H.Y.C.); jicsam@ummc.edu.my (I.-C.S.); 2Department of Microbiology and Molecular Medicine, University of Geneva, 1211 Geneva, Switzerland; han.tee@unige.ch; 3Department of Biomedical Sciences, Faculty of Medicine, Universiti Malaya, Kuala Lumpur 50603, Malaysia; kcong@um.edu.my; 4Comparative Medicine and Technology Unit, Institute of Bioscience, Universiti Putra Malaysia, Serdang 43400, Malaysia; kartini.jasni@upm.edu.my; 5Department of Biomedical Sciences, Faculty of Medicine and Health Sciences, Universiti Putra Malaysia, Serdang 43400, Malaysia; syahril@upm.edu.my or; 6Malaysia Genome and Vaccine Institute, National Institutes of Biotechnology Malaysia, Kajang 43000, Malaysia

**Keywords:** enterovirus A71, coxsackievirus A16, hand–foot–mouth disease, recombination, viral virulence, innate

## Abstract

**Background/Objectives**: Enterovirus A71 (EV-A71) and coxsackievirus A16 (CVA16) are major causative agents of hand, foot and mouth disease (HFMD), often co-circulating and occasionally undergoing genetic recombination. While natural recombinants often involve genomic regions encoding non-structural proteins, their effects on replication and pathogenesis remain unclear. **Methods**: To address this, four chimera viruses (Chi-CCE, Chi-ECE, Chi-EEC, and Chi-CEC) were constructed with 5′UTR, capsid P1, and non-structural P2 and P3 genes, from CVA16 (denoted as C) or EV-A71 (denoted as E). These chimeras were tested for replication kinetics and cytopathic effects in rhabdomyosarcoma cells while *in vivo* virulence and protection efficacy were evaluated using a newborn BALB/c mouse model. **Results**: All chimeric viruses remained viable and exhibited higher replication than CVA16. *In vivo*, all chimeric viruses were avirulent except Chi-CCE and CVA16, which showed high virulence and viral titres in the brains and limbs of infected newborn mice. This suggests that 5′UTR and capsid P1 genes of CVA16 are critical genetic determinants of virulence. Notably, only the anti-inflammatory cytokine IL-10 was elevated, suggesting potential immune modulation during infection. Inactivated Chi-CCE immunisation conferred 100% protection against lethal CVA16 or mouse-adapted EV-A71 challenge revealing its potential as a bivalent vaccine candidate. **Conclusions**: Our study demonstrates that recombination between CVA16 and EV-A71 influences viral virulence and protection efficacy with implications for future development of multivalent vaccines.

## 1. Introduction

*Enteroviruses alphacoxsackie* including enterovirus A71 (EV-A71) and coxsackievirus A16 (CVA16), are major causative agents of hand, foot and mouth disease (HFMD) affecting children aged 5 years old and below. These viruses share a high degree of genome sequence similarity, with an average 78.5% nucleotide identity and 87% amino acid similarity, yet differ significantly in pathogenicity across human and mice models [1,2]. EV-A71 is more frequently linked to neurological complications in humans, whereas CVA16 demonstrates higher virulence in mice.

Their genomes are organised into three key regions: the 5′ untranslated region (5′ UTR), a single open reading frame (ORF) which encodes both structural and non-structural proteins, and the 3′ untranslated region (3′ UTR). The 5′ UTR has a cloverleaf-like structure essential for viral RNA synthesis and serves as an internal ribosomal entry site (IRES) to mediate cap-independent translation [3,4,5]. The single ORF is cleaved into four structural proteins (VP1–VP4) and seven non-structural proteins (2A–2C and 3A–3D). Structural proteins determine host receptor specificity and antigenicity [6,7,8], while non-structural proteins regulate viral replication, polyprotein processing, and immune evasion [9,10,11,12].

Genetic recombination is a prevalent and influential mechanism in the evolution of enteroviruses (EVs), contributing to their genetic diversity and adaptability. Recombination events frequently occur within the 5′UTR and non-structural protein coding region [13,14,15,16,17,18,19,20,21]. These events can involve genetic exchanges within a single EV type (intratypic recombination) or across species (intertypic recombination). Recombination can shape the phylogenetic landscape of the EV genus, leading to the emergence of distinct species and evolutionary divergence of EV [14]. Within each human EV species, the presence of incongruent phylogenetic tree topologies across UTR, structural, and non-structural regions reveals frequent occurrence of intra- and intertypic recombination. Phylogenetic analysis revealed that recombination clustered current human EV 5′ UTR sequences into two distinct groups: Group I, consisting of EV-C and EV-D, and Group II, comprising EV-A and EV-B [14].

During HFMD outbreaks, frequent co-circulation of CVA16 and EV-A71 increases the likelihood of genetic exchange, leading to recombinant strains with altered fitness, antigenicity, and pathogenicity [22]. Specific hotspot recombination breakpoints have been identified between CVA16 and EV-A71, notably within the 5′UTR, 2A, and 3D^pol^ regions [15,16,23,24]. These recombination events are not random but occur at these hotspots which are more conserved across different enteroviruses. This modular exchange of genetic material is a key driver of EV speciation and ongoing evolution, facilitating the emergence of novel strains.

While recombination has been documented, the functional impact of structural and non-structural protein exchange between these two closely related enteroviruses remains incompletely understood. In this study, we constructed chimeric viruses with different structural and non-structural genes of CVA16 and EV-A71 to assess their phenotypic properties in rhabdomyosarcoma (RD) cells and virulence in a mouse model. Our findings offer insights into how genomic recombination shapes enterovirus pathogenicity, viral fitness, and immune modulation and may impact rational vaccine design and antiviral development.

## 2. Materials and Methods

### 2.1. Cell Lines and Viruses

RD cells (ATCC no: CCL-136) were propagated in Dulbecco’s modified Eagle’s medium (DMEM, Gibco, Waltham, MA, USA) containing 10% fetal bovine serum (FBS) under standard culture conditions of 37 °C with 5% CO_2_. For viral studies, we used infectious clones: pCMV-CVA16 derived from CVA16 strain PM-22159-02 (GenBank JQ746673) and pCMV-EV-A71 derived from EV-A71 strain 5865/sin/000009 (subgenotype B4; GenBank AF316321) [2,25]. In this work, we were unable to rescue the initial CVA16; however, we obtained the viruses after substituting the 3′UTR with that of EV-A71. The gene sequence is still 99.8% identical to the CVA16 genome. No significant differences in replication kinetics were observed between the CVA16 clinical isolate PM-22159-02 and its corresponding infectious clone *in vitro*. A mouse-adapted EV-A71 (MP4) (BEI Resources, catalogue number NR-472) was used for lethal challenge in this study as the EV-A71 strain 5865/sin/000009 carrying the VP1-145Q is non-lethal in mice [26,27].

### 2.2. Construction of Chimeric Viruses

Four chimeric plasmids (pCMV-Chi-CCE, pCMV-Chi-ECE, pCMV-Chi-EEC, and pCMV-Chi-CEC) were constructed with different 5′UTR, P1, P2, and P3 regions from CVA16 (denoted as C) and EV-A71 (denoted as E). The nomenclature of the chimeric viruses is according to whether the 5′UTR, structural (P1), and non-structural (P2 and P3) regions are from CVA16 or EV-A71; for example, Chi-CCE contains 5′UTR and P1 regions from CVA16 and P2 and P3 regions from EV-A71. CVA16 and all chimeras have 3′UTR of EV-A71 except Chi-CEC. Briefly, the CVA16 and EVA71 structural and non-structural protein genomic sequences were amplified from pCMV-CVA16 and pCMV-EV-A71 by exponential megapriming polymerase chain reaction (EMP) using Q5 High-Fidelity DNA polymerase (NEB, Ipswich, MA, USA). PCR was performed with 100 ng of plasmid template, 1 × Q5 reaction buffer, 0.02 U/µL Q5 High Fidelity DNA polymerase, 200 µM dNTPs, and 0.5 µM of primers (Appendix A) following the parameters 98 °C for 30 s, 35 cycles at 98 °C for 10 s, 66 °C/72 °C for 30 s, and 72 °C for 2 min followed by 72 °C for 2 min. The PCR products were purified and ligated into pCMV vectors resulting in pCMV-Chi-CCE, pCMV-Chi-ECE, pCMV-Chi-EEC, and pCMV-Chi-CEC before transforming into *E. coli* XL10-Gold-competent cells. Positive colonies grown on Luria-Bertani (LB) agar with kanamycin were screened using colony PCR and DNA sequencing with primers (Appendix A). Confirmed colonies were cultured in 50 mL LB broth containing 5 µL/mL kanamycin, and plasmids were extracted using the PureLink Plasmid Midiprep Kit (Invitrogen, Waltham, MA, USA).

### 2.3. Virus Propagation, Titration, and Purification

For transfection, RD cells (4 × 10^5^ cells/well) in 6-well plates were loaded with 2 µg of purified plasmids using Lipofectamine LTX Reagent with PLUS Reagent and incubated for 4 h at 37 °C (Invitrogen). Then, the transfection mixture was replaced with 10% FBS DMEM. The EV-A71 and chimeric viruses were harvested when 80% of the cells showed cytopathic effect (CPE). CVA16, which does not form cytopathic effect at the P0 stage, was collected at 5 days post-transfection. All viruses were harvested by three freeze–thaw cycles for the initial P0 virus stock. These viruses were then propagated to obtain P1 virus stock for downstream experiments. All viruses were verified by sequencing prior to use. The viral supernatants were concentrated, desalted, and purified using a Centriprep 30 kDa cutoff column filter (Merck, Darmstadt, Germany) through centrifugation at 1500× *g* for 15 min. The filtered samples then underwent high-speed centrifugation at 40,000× *g* at 4 °C (Sorvall, Thermofisher, Waltham, MA, USA) and the resulting supernatant was further concentrated to 10% of its original volume via ultracentrifugation. To perform ultracentrifugation, 2 mL of a solution containing 30% sucrose pH 7.5 and 13.5 mL of the filtered supernatant was added and centrifuged at 125,000× *g* for 4 h at a 4 °C in an Optima XPN ultracentrifuge (Beckman Coulter, CA, USA). After ultracentrifugation, the supernatants were discarded, and the pellets were resuspended in 500 µL of phosphate-buffered saline (PBS).

### 2.4. In Vitro Replication Kinetics and Plaque Assay

The viruses were inoculated at a multiplicity of infection (MOI) of 0.1 into 7.0 × 10^4^ RD cells/well in a 24-well plate. After an hour of incubation, the inoculum was replaced with 2% FBS DMEM. Subsequently, the virus-infected cells were incubated at 37 °C with 5% CO_2_ and harvested at 24, 48, 72, 96, and 120 hours post-infection (hpi) for virus titration. Median tissue culture infectious dose (TCID_50_) was performed to titrate the virus while a plaque assay was performed to compare plaque morphology between the EV-A71, CVA16, and chimeric viruses. The TCID_50_ of the viruses were calculated using the Reed–Muench method. RD cells (2.5 × 10^5^ cells per well) in a 12-well plate were inoculated with 500 µL of serially diluted virus, incubated for 1 h at 37 °C, and overlaid with plaque media (0.8% carboxymethylcellulose agar, DMEM, and 2% FBS). After 5 days at 37 °C in 5% CO_2_, the cells were fixed with 3.7% formaldehyde, stained with crystal violet, and rinsed. The plaques were observed, and their diameters were measured.

### 2.5. Mice Experiments

Mouse experiments were approved by the Institutional Animal Care and Use Committee (UPM/IACUC/AUP-R042/2022) and conducted at the Comparative Medicine and Technology Unit, Institute of Bioscience, Universiti Putra Malaysia. Special pathogen-free BALB/c mice (6–8 weeks old) were obtained from the Malaysian Institute of Pharmaceuticals and Nutraceuticals, National Institutes of Biotechnology Malaysia (iPharm—NIBM, Penang, Malaysia) and housed in individually ventilated cages. Newborn mice were selected due to their high susceptibility to enterovirus infection [28,29,30]. Body weight and clinical signs were monitored every 2–3 days. Disease severity was graded on a scale of 0–5 as below: 0, healthy; 1, reduced mobility; 2, limb weakness; 3, one limb paralysis; 4, two limbs paralysis; and 5, death [26]. Based on previous experiments, the median lethal dose (LD_50_) values for CVA16 and MP4 were determined to be 0.034 TCID_50_ and 10^3^ TCID_50_, respectively [2,31].

### 2.6. Determination of Virulence of Chimeric Viruses

To investigate the virulence of the chimeric viruses, newborn mice (*n* = 5) were intracerebrally (i.c.) inoculated with chimeric viruses at a dosage of 9.0 × 10^3^ pfu/mouse, which represents the maximum viral dosage to ensure that potential neurovirulence was not compromised by an insufficient viral dosage [32,33]. Mice inoculated with PBS and the same dosage of the CVA16 and EV-A71 viruses were included as controls. The body weights and survival rates of the mice were monitored for 14 days post-infection (dpi). The viral titre of brains and limbs was determined using a TCID_50_ assay [31].

Cytokines including interferon-γ (IFN-γ), interleukin-2 (IL-2), IL-4, and IL-10 induced during virus infection in mice were measured using a mouse Th1/Th2 enzyme-linked immunosorbent assay (ELISA) kit (Invitrogen) [2]. Total lysates were collected from infected mouse organs to measure cytokine production. Mouse brains and limbs were homogenised at 4000 rpm for 30 s in a 1:1 weight-to-volume ratio serum-free media using 2.8 mm Omni ceramic beads. Supernatants were collected and stored for further analysis. Briefly, after coating and blocking the plate, 50 µL of 1:2 diluted supernatant, standards, and ELISA diluent (as blank) were added to the wells in replicates and incubated for 2 h, followed by incubation with biotin-conjugated anti-mouse antibodies for 1 h. Streptavidin-HRP was added to each well and incubated for 30 min, followed by the addition of tetramethylbenzidine substrate solution and a 15 min incubation. After adding the stop solution, the plate was read at 450 nm.

### 2.7. Protective Effect of Inactivated Chimeric Viruses Against Lethal Virus Challenge

Due to the high lethality of live CVA16 in newborn mice even at very low dosages [2,34,35,36,37], inactivated viruses were used. Following viral titration using a TCID_50_ assay, the EV-A71, CVA16, and chimeric viruses were inactivated with 0.25% (*v*/*v*) formaldehyde for 7 days at 37 °C to produce iEV-A71, iCVA16, iChi-CCE, iChi-ECE, iChi-EEC, and iChi-CEC. Prior to inactivation, the supernatant was filtered to remove virus aggregates that could reduce formaldehyde penetration into the virus particles. The absence of CPE in RD cells indicates no live virus and confirms the effectiveness of formaldehyde inactivation.

Groups of newborn BALB/c mice (*n* = 5) were immunised via the intraperitoneal (i.p.) route with the inactivated EV-A71, CVA16, or chimeric viruses on day −7 at dosages of 1.0–5.0 × 10^5^ pfu/mouse. Additional booster doses with a similar dosage were i.p. inoculated on day -2 for mice receiving the MP4 challenge at 5 LD_50_ on day 0, as a single dose of iEV-A71 virus immunisation was found to be insufficient to provide protection against MP4 virus infection [31]. Another group of mice were challenged with 5 LD_50_ of CVA16 on day 0 without a booster dose. Over the 14-day period, the infected mice were observed for any clinical signs and weight loss. Survival of the infected mice was recorded to examine the protective effect of immunisation with chimeric viruses upon lethal CVA16 and MP4 challenge.

The brains and limbs were collected for virus titration using a TCID_50_ assay, reverse transcription quantitative real-time PCR (RT-qPCR) quantification, and histology analysis. The supernatant from the homogenised tissues was processed with viral RNA using the QIAamp Viral RNA Mini Kit (Qiagen, Hilden, Germany) following the manufacturer’s instructions. Then, RT-qPCR for CVA16 and MP4 were carried out with respective gene-specific primers and TaqMan probes using the LightCycler Multiplex RNA Virus Master (Roche, Basel, Switzerland). The cycling protocol started with an incubation cycle of 50 °C for 10 min, 40 cycles at 95 °C for 20 s, and 60 °C for 30 s, followed by a holding cycle at 40 °C for 30 s [31].

### 2.8. Histopathology and Immunohistochemistry Staining

Limb and brain tissues were collected from severely infected mice at 7 dpi and from surviving mice at 14 dpi. After fixing in 10% neutral buffered formalin for at least seven days, paraffin-embedded samples were sectioned at 4 μm thickness and stained with hematoxylin and eosin (H&E) for histological examination. A modified immunohistochemistry (IHC) protocol was employed to detect CVA16 and EV-A71 antigens [38]. A polyclonal antibody recognising the N-terminal VP1 peptide (GDRVADVIESSIGDC) was used at dilution 1:5000 for MP4 and 1:2000 for CVA16 detection. The samples were then incubated with a horseradish peroxidase (HRP)-conjugated anti-rabbit secondary antibody (Dako, Santa Clara, CA, USA) with 3,3′-diaminobenzidine tetrahydrochloride (DAB) as the substrate.

### 2.9. Statistical Analysis

Statistical analysis was performed using GraphPad Prism version 9 (GraphPad Software, San Diego, CA, USA). Viral titres in the RD cells and infected animals were compared using a two-way analysis of variance (ANOVA) with Bonferroni post hoc tests. The Mantel–Cox log-rank and Gehan–Breslow–Wilcoxon tests were used to analyse the survival rates of infected mice. Differences in viral titres, RNA copy numbers, and cytokine concentrations were assessed by one-way ANOVA followed by Bonferroni correction for multiple comparisons. The data are shown as the mean ± SD, and a threshold of *p* < 0.05 was used to indicate statistical significance.

## 3. Results

### 3.1. CV-A16 Chimeric Viruses Were Viable with Enhanced Fitness

Chimeric viruses were constructed with different gene segments from CVA16 and EV-A71 (Figure 1A). The replication kinetics of chimeric viruses were compared to CVA16 and EV-A71 viruses. RD cells were infected with EV-A71, CVA16, and chimeric viruses at an MOI of 0.1 and harvested at different time points (24 to 120 hpi) for virus titration (Figure 1B). CVA16 showed the lowest replication efficiency, peaking at a titre of approximately 6 log_10_ TCID_50_/mL around 24 hpi and then steadily declining. All chimeric viruses (Chi-CCE, Chi-ECE, Chi-EEC, and Chi-CEC) exhibited higher viral titres than CVA16 throughout the time course, particularly at 24–72 hpi. EV-A71 reached the highest peak titre (~8 log_10_ TCID_50_/mL) at 24 hpi and maintained it until 48–72 hpi before gradually decreasing. Similar to EV-A71 virus-infected cells, all chimeric virus-infected cells showed strong CPE at 72 h post-infection indicating that these chimeric viruses were viable (Figure 1B). This indicates that replacing structural and/or non-structural proteins of either CVA16 or EV-A71 alter the replication efficiency in RD cells. A plaque assay was performed, and the plaque sizes were measured. The CVA16 virus rescued from pCMV-CVA16 did not lead to production of visible plaques. Chi-CEC and Chi-CCE viruses displayed comparable plaques with mean plaque sizes of 0.90–0.95 mm, similar to the EV-A71. Taken together, introduction of any part of the EV-A71 genes into the CVA16 genome enhanced virus fitness, leading to comparable plaque phenotypes as EV-A71.

### 3.2. Viral Virulence of Chimeric Viruses in Newborn Mice

To examine the virulence of chimeric viruses with substituted CVA16 or EV-A71 structural proteins or non-structural proteins, groups of newborn mice were inoculated i.c. with EV-A71, CVA16, or chimeric viruses at a dosage of 9 × 10^3^ pfu/mouse (Figure 2A).

While all EV-A71-infected mice showed retarded growth without paralysis, all mice infected with CVA16 showed severe clinical symptoms and died at 7 dpi (Figure 2B). Notably, Chi-CCE with both the 5′UTR and P1 region from CV-A16 displayed high virulence similar to CVA16, with all infected mice dying by 9 dpi. Surprisingly Chi-ECE with only the CV-A16 P1 region substituted into an EV-A71 genome displayed attenuated virulence with no deaths observed. This finding implies that the 5′UTR compatibility with the P1 region is essential and may act on the CVA16 P1 region in determining the viral virulence of CVA16. In addition, all mice infected with chimeric viruses Chi-CEC and Chi-EEC survived, indicating that the substitutions in the UTR and P2/P3 regions of the EV-A71 genome resulted in the same virulence phenotype as EV-A71.

To further investigate the effects of viral replication on tissue tropism and pathogenicity during the infection, brains and limbs were collected from mice with severe clinical symptoms. Both the CVA16 and highly virulent Chi-CCE virus exhibited similar viral titres in the brains, measuring approximately 3.5 log_10_ TCID_50_/mL (Figure 2C). Chi-CCE-infected mice showed a higher viral titre in the limbs at 3.7 log_10_ TCID_50_/mL compared to the CVA16 titre of 2.3 log_10_ TCID_50_/mL (Figure 2C). Real-time qPCR also revealed similar viral RNA loads for both virus-infected limbs and brains ranging from 6 to 7.5 log_10_ copies/µL (Figure 2C). This indicates that active viral replication of Chi-CCE and CVA16 contributed to the high mortality rate in mice.

PBS-inoculated mice showed no inflammation or viral antigen in their skeletal muscle (Figure 2D; Table 1). However, both CVA16- and Chi-CCE-infected mice showed severe necrosis and mild myositis (arrowheads, Figure 2D; Table 1) with an abundance of IHC-positive tissues in the skeletal muscles (arrows, Figure 2D; Table 1) correlated with the viral load and high mortality. Although viral titres were detected in the brains of CVA16- and Chi-CCE-infected mice, neither antigens nor histopathological changes were observed in the brain areas, including the cerebral cortex, pons, midbrain, and cerebellum (Table 1). This discrepancy may be attributed to the limited sensitivity of IHC potentially resulting in lower detection of viral antigens. Alternatively, the detected viral RNA by qPCR could be due to the residual blood in the brain either from systemic viremia following peripheral muscle infection or from the intracranial injection itself, consistent with known viral tropism of CVA16 in muscle tissue [38].

To assess the cellular immune responses triggered by CVA16 and Chi-CCE infections, the levels of Th1-type (IFN-γ and IL-2) and Th2-type (IL-4 and IL-10) cytokines in the limbs and brains were examined using ELISA. IFN-γ and IL-4 were not significantly elevated in mice infected with CVA16 or Chi-CCE viruses when compared to mice inoculated with PBS (Figure 2E). There was no significant difference in IL-2 production in the limbs, but lower production (100 pg/mL) was observed in the brains of mice infected with CVA16. Both CVA16 and Chi-CCE infections resulted in increased IL-10 in the limbs (1200–1800 pg/mL) and brains (670–1200 pg/mL) (Figure 2E). Both elevated cytokine profiles suggest an anti-inflammatory response during CVA16 and Chi-CCE infections.

### 3.3. iChi-CCE Confers Protection Against CVA16 Challenge

To evaluate the protective efficacy of chimeric viruses as potential vaccines against lethal CV-A16 challenge, we generated inactivated forms of Chi-CCE and Chi-ECE by formaldehyde inactivation (termed iChi-CCE and iChi-ECE, respectively), alongside inactivated iCV-A16 and iEV-A71. These inactivated viruses were inoculated into newborn mice prior to CVA16 challenge (Figure 3A). Upon CVA16 challenge, iCVA16- or iChi-CCE- immunised mice resulted in full protection with 100% survival (Figure 3B). In contrast, iEV-A71 demonstrated partial heterologous protection, resulting in 80% survival, while iChi-ECE exhibited a diminished protection level with a survival rate of 60%.

Viral titration in the limbs and brains revealed consistently low viral titres at around 0–2 log_10_ TCID_50_/mL across all groups regardless of severity of virus infection (Figure 3C). Reduced CVA16 infection in iCVA16-, iChi-CCE-, and iEV-A71-immunised mice was supported by RT-qPCR analysis with the presence of low viral RNA levels in both the limbs and brains (3–4 log_10_ RNA copies) (Figure 3D). In contrast, dead mice from the PBS and iChi-ECE inoculation exhibited markedly elevated viral RNA in their limbs (up to 9 log_10_ RNA copies) and brains (up to 7 log_10_ RNA copies) (Figure 3D). RT-qPCR confirmed lower viral RNA levels in survivor mice, with significantly higher levels detected in the PBS and iChi-ECE groups, correlated with mice mortality. In the CVA16 challenge group, PBS-inoculated mice showed severe infiltration of immune cells and abundant CVA16 viral antigen in skeletal muscle (arrows and arrowheads, Figure 3E; Table 2). Mice with iCVA16, iChi-CCE, and iEV-A71 immunisation showed no signs of inflammation and no detectable CVA16 viral antigens in the skeletal muscle, correlated with no infection and strong protection (Figure 3F,G,I; Table 2). iChi-ECE-immunised mice showed inflammatory cell infiltration and an abundance of viral antigen-positive muscle fibres (arrows and arrowheads, Figure 3H; Table 2), which correlated with the higher virus titre and viral RNA loads detected (Figure 3C). Additionally, all the brains collected from the infected mice did not show any degree of inflammation, abnormality, and viral antigens (Table 2). The overall histology findings corroborated mice survival.

### 3.4. iChi-CCE Confers Protection Against MP4 Challenge

The protective efficacy of chimera virus immunisation was also tested against lethal challenge with the mouse-adapted EV-A71 MP4 strain (Figure 4A). In this group, mice received double doses of inactivated virus immunisation, since our previous study demonstrated that a single dose of iEV-A71 was insufficient to confer full protection against MP4 challenge [31]. Newborn mice immunised with iChi-CCE had 100% survival upon MP4 challenge similar to those with iEV-A71 immunisation (Figure 4B). In comparison, iChi-ECE and iCVA16 immunisation provided partial protection, with survival rates of 75% (Figure 4B). Consistent with the survival rates, mice that survived MP4 infection had lower viral titres in both limbs and brains (0–4 log_10_ TCID_50_/mL for both) (Figure 4C), as well as reduced viral RNA copies ranging from 1 to 3 log_10_ RNA copies in both limbs and brains (Figure 4D). An exception was observed in one limb sample from the iCVA16-immunised group which showed a higher MP4 viral titre of 5 log_10_ TCID_50_/mL (Figure 4C). Consistently, the deceased mice from the PBS-inoculated group and iChi-ECE-inoculated group exhibited high viral titres in the limbs (4–6 log_10_ TCID_50_/mL) and brains (0–5 log_10_ TCID_50_/mL), along with an elevated viral RNA load in the limbs (2–8 log_10_ RNA copies) and brains (2–6 log_10_ RNA copies). Interestingly, higher viral titres were observed in the PBS-inoculated and iChi-EEC-immunised limbs and brains as compared to RT-qPCR analysis of viral RNA loads. Overall, mice survival correlated with both lower viral titres and viral RNA loads.

In the histopathology analysis, the PBS-inoculated mice showed infiltration of immune cells and the presence of the MP4 viral antigen (arrows and arrowheads, Figure 4E; Table 2). The iChi-CCE-immunised mice showed an absence of immune cell infiltration and MP4 viral antigen as observed in the iEV-A71 immunised mice (Figure 4G,I; Table 2). Meanwhile, the iCVA16-immunised mice which developed severe signs on 7 dpi upon MP4 challenge showed infiltration of immune cells and an abundance of MP4 antigens (Figure 4F; Table 2). The iChi-ECE-immunised mice demonstrated relatively lower levels of immune cell infiltration and the presence of viral antigen as compared to the PBS-inoculated mice (Figure 4H; Table 2). Taken together, Chi-CCE immunisation clearly provided protection against both CVA16 and MP4 with no visible inflammation.

## 4. Discussion

Recombination is a natural process in which a virus acquires a mosaic genome or exchanges genetic sequences, potentially with altered viral replication, viral growth, and virulence compared to its parental viruses. While intratypic recombination of EV-A71 has been experimentally demonstrated in cell-based assays [24,40], evidence for intertypic recombination remains limited and is primarily supported by phylogenetic analysis. According to Santti et al., recombination in 5′UTR resulted in Group I enteroviruses (including poliovirus) and Group II enteroviruses (including EV-A71 and CVA16) [14]. The 5′UTR differences can influence translation efficiency, replication capacity, and ultimately virulence. Further recombination in the coding regions speciate the Group I enteroviruses into clusters C and D, while Group II enteroviruses fall into clusters A and B. Comparative analyses of enterovirus strains suggest that much of the variation within clusters arises from natural intraspecies recombination, frequently occurring at the P1/P2 junction. This is directly relevant to our study design, as our chimeric constructs were generated with breakpoints at similar junctions (particularly at the P1/P2 boundary) to dissect the respective roles of the 5′UTR, P1, and P2/P3 in determining viral phenotype. To study recombination in cluster A enteroviruses, we characterised chimeric viruses derived from CVA16 and EV-A71 as an example of intertypic recombination to examine the impact of structural and non-structural protein exchanges on viral replication, pathogenesis, and immunogenicity. Based on findings from a poliovirus trans-complementation study, which showed that long non-structural precursors particularly the P3 non-structural genes are essential for rescuing replication-defective mutants *in vivo*, we designed our chimeric construct by replacing the entire P2 and P3 region of CVA16 with that of EV-A71 [41]. The successful rescue of these constructs further reaffirmed the importance of conservation of non-structural proteins, ensuring precise precursor processing and functional replication complex formation.

This study showed that chimeric virus Chi-CCE resulted in high virulence when administered to mice. This phenotype closely resembled that of infection with the CVA16 strain, suggesting that substitution of the non-structural region from EV-A71 was insufficient to attenuate virulence of the recombinant. To mitigate the virulence of Chi-CCE, the virus was inactivated. Immunisation with iChi-CCE conferred full protection against both CVA16 and MP4 challenges, similar to the effect of immunisation with iCVA16 (against CVA16) and iEV-A71 (against MP4). In addition to its protective efficacy, iChi-CCE influenced host immune responses, as shown by the upregulation of IL-10 accompanied by low IFN-γ expression. IL-10 can act as an anti-inflammatory or pro-inflammatory cytokine. The increase in IL-10 without significant changes in IFN-γ or IL-2 (both pro-inflammatory) suggests that CVA16/Chi-CCE may suppress excessive inflammation to reduce tissue damage. Similar immune modulation patterns have been observed in EV-A71-infected or -immunised mice with elevated IL-10 peaking around 4 dpi and reduced or unchanged IFN-γ [42]. Some studies reported simultaneous increases in IL-10 and IFN-γ in serum [28,43], but the discrepancy is likely attributed to differences in sampling timepoints. Overall, iChi-CCE demonstrated robust cross-protection, highlighting its potential as a promising vaccine candidate against HFMD.

Our study explored how different genomic regions influence viral fitness and phenotypic traits and showed that incorporation of any EV-A71 genetic segments into CVA16 enhances viral fitness and modifies its phenotypic characteristics in RD cells. This suggests both UTRs and non-structural regions from EV-A71 contribute functionally to replication efficiency and cellular tropism. In contrast, the introduction of CV-A16 genetic segments into EV-A71 did not affect virus fitness. Previous studies have demonstrated that swapping combinations of the 5′UTR and P2 P3/3′UTR from poliovirus Sabin 2 with those of non-polio EV *coxsackiepol* viruses produced viable recombinant viruses capable of inducing cell lysis, highlighting cross-species compatibility among enteroviral gene regions [44]. However, compatibility among non-structural proteins varies widely across enteroviruses. For example, replacement of EV-A71 with those from EV *alphacoxsackie* viruses such as CVA6, CVA16, EV-A89, EV-A91, and EV-A125 resulted in both replication-competent and replication-defective recombinant viruses [45]. Another study with genetic exchange of 5′UTR-P1-2A or 2BC-P3-3′UTR also demonstrated generation of viable recombinant viruses with comparable replication kinetics, indicating functional compatibility of these genomic regions through recombination [46]. Also, a recombinant of rhinovirus A16 with the EV-A71 5′UTR exhibited reduced replication efficiency, suggesting species-specific UTR compatibility is still required for virus fitness [47]. These findings suggest that functional compatibility plays a key role in determining the success of recombinant viruses but not just the sequence similarity.

To further explore the role of 5′UTR, iChi-ECE with EV-A71 5′UTR was included in this study. This construct showed lower protection toward CVA16 and MP4, emphasising the importance of both the 5′UTR and P1 region in generating effective protective immunity. Substitution of 5′UTR alone has frequently been reported to be sufficient to alter viral virulence in mice. For instance, replacement of the 5′UTR of poliovirus Sabin 2 with that of a non-polio Enterovirus *coxsackiepol* strain as well as a recombinant between MP4 and CVA16 led to significantly increased pathogenicity *in vivo* [44,48]. Similarly, replacement of the 5′UTR or 3′UTR of a virulent EV-A71 strain with that of a mild strain further confirmed the role of the 5′ UTR in viral pathogenicity with reduced disease severity [49]. The critical role of the 5′UTR, particularly the IRES in determining viral pathogenicity and neurotropism, has been demonstrated [50]. Five unique substitutions in the 5′UTR among 21 genomic changes identified in acute flaccid myelitis-associated EV-D68 strains have also been found in other neurovirulent enteroviruses, including EV-D70, poliovirus, and EV-A71, and this may be a potential conserved mechanism underlying neurovirulence [51]. An additional short upstream open reading frame (uORF) encoding a small virus protein of <10 kDa was recently discovered between the 5′UTR and VP4 of enteroviruses [52]. Although not essential for viral replication, this uORF protein has been correlated with viral invasion by facilitating virus particle release from vesicles. Therefore, 5′UTR substitutions in this study may affect both IRES activity and uORF translation, potentially altering ribosome recognition and downstream capsid protein expression. Since the uORF product facilitates viral release, disrupting its translation may reduce pathogenicity or shifted tissue tropism. Combined with capsid changes like VP1 loop mutations, these effects may interact synergistically or antagonistically, influencing viral fitness, immune evasion, and neurotropism. Studies have shown that swapping capsid components such as VP1 loops or the SP70 epitope between CVA16 and EV-A71 can elicit broad immunity [2,53,54]. Similarly, recombination between coxsackievirus A9 and CVB3 demonstrated that capsid proteins determine viral growth, phenotype, and immunogenicity [55,56].

Our study has some key limitations. This study only investigates recombination in structural and non-structural genes. In our study, the 5′UTR and P1 of CVA16 (Chi-CCE) showed high virulence while conferring high protection, whereas CVA16 P1 alone (Chi-ECE) was insufficient to confer high protection against lethal virus infection. However, the lack of 5′UTR-only chimera constructs Chi-CEE and Chi-ECC in this study restricts our ability to assess whether the observed protective effect results from the independent effect of the 5′UTR or from its interaction with the P1 region. Future studies should incorporate such constructs to better dissect the role of 5′UTR in governing viral replication, virulence, and host immune responses. A well-known example highlighting the impact of the 5′UTR is how the nucleotide position 472 in poliovirus (Group I) attenuates viral replication in neuronal tissues and serves as the genetic basis of the live attenuated Sabin vaccine [57]. This illustrates how 5′UTR changes can strongly affect viral fitness and virulence, consistent with the rationale of our chimeric virus approach.

For future development of iChi-CCE as a vaccine candidate, alternative animal models, such as hSCARB2 transgenic mice which more closely mimic human infection could be used, and comprehensive immunological analysis including neutralising antibody titres, T-cell responses, and cytokine profiles will be necessary to elucidate the mechanisms of Chi-CCE-induced immune protection.

The use of one single viral backbone as a candidate HFMD vaccine eliminates the need for two separate viruses, simplifies production, and lowers manufacturing complexity and cost. Its use as a vaccine candidate is well positioned for resource-limited settings, where cost, stability, and ease of deployment are critical.

## 5. Conclusions

Our study demonstrates that recombination between the CVA16 and EV-A71 genomic regions modulates viral virulence and protective efficacy, with important implications for the design of future multivalent vaccines to combat HFMD. Specifically, we highlight the roles of the 5′UTR and P1 structural gene region in shaping viral pathogenicity.

## Figures and Tables

**Figure 1 vaccines-13-01017-f001:**
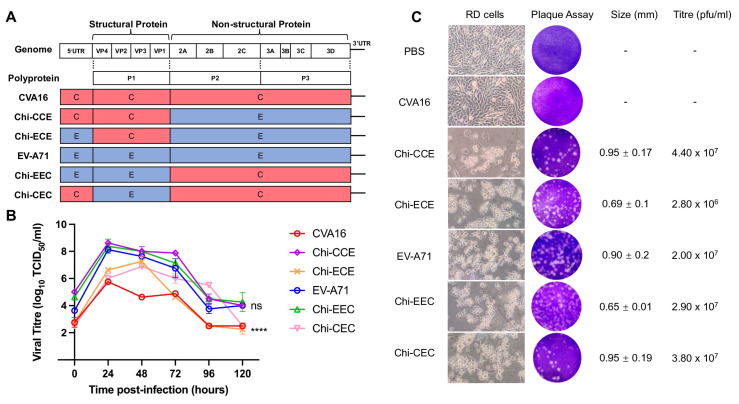
Construction and phenotypic characterisation of EV-A71, CVA16, and chimeric viruses. (**A**) Construction of chimeric viruses by replacing the 5′UTR, P1, P2, and P3 regions with genes from either CVA16 (red) or EV-A71 (blue). (**B**) Replication kinetics of chimeric viruses at 0, 24, 48, 72, 96, and 120 hpi and titrated using TCID_50_. Results are presented as mean ± SD (*n* = 2). Error bars indicate standard deviations from the duplicates. **** *p* < 0.0001. (**C**) RD cells were infected with viruses at MOI = 0.1 and characterisation of EV-A71, CVA16, and chimeric viruses based on CPE formation and plaque assays were performed. Plaque assay analysis revealed the plaque morphology, plaque size, and the viral titre at 3 dpi.

**Figure 2 vaccines-13-01017-f002:**
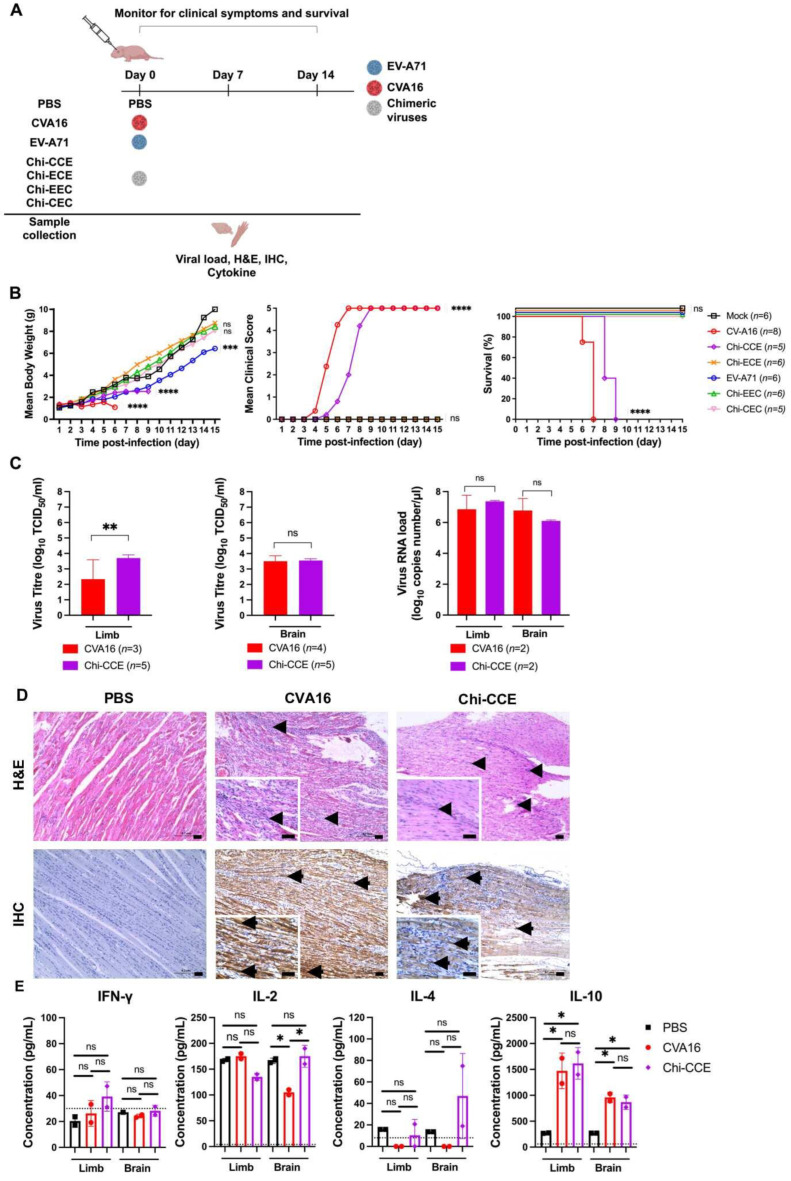
Virulence of chimeric viruses in mice. (**A**) Diagram illustrating the *in vivo* experimental design used to assess chimeric virus virulence. (**B**) Mean body weight, clinical score, and survival rate of infected mice were monitored, with the number of mice indicated as *n*. Chimeric virus-infected mice were compared with PBS-inoculated mice. (**C**) Viral titration (log_10_ TCID_50_/mL) using TCID_50_ assay and virus RNA load (log_10_ copies/µL) using real-time qPCR in brains and limbs. The data are presented as mean ± SD (*n* > 2). Viral titre and viral RNA load in Chi-CCE-infected mice were compared with CVA16-infected mice. (**D**) H&E staining and IHC analysis for PBS-inoculated, CVA16-infected, and Chi-CCE-infected mice. Arrowheads indicate inflammatory cell infiltrates, while arrows indicate viral antigens. Scale bar: 50 µm. Magnification: 10× and 40× (small box). (**E**) The concentrations of cytokines including IFN-γ, IL-2, IL-4, and IL-10 in the sample supernatants were measured by ELISA. Dotted lines indicate limit of detection for the ELISA. Results are presented as mean ± SD (*n* = 2). Statistically significant differences after Bonferroni correction are denoted with * *p* < 0.05, ** *p* < 0.01, *** *p* < 0.001, **** *p* < 0.0001, and no significant (ns).

**Figure 3 vaccines-13-01017-f003:**
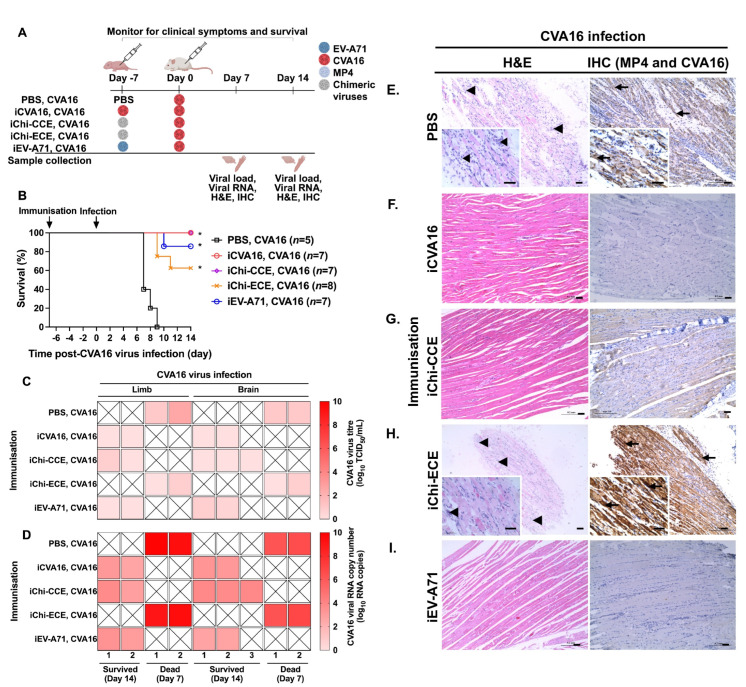
Protective effects of inactivated chimeric viruses against CVA16 virus challenge. (**A**) Schematic of *in vivo* experiments to examine the virulence of chimeric viruses. (**B**) Survival percentage of infected mice were recorded. Number of mice are denoted as *n*. Chimeric virus-immunised mice were compared with PBS-inoculated mice. (**C**) Viral titration (log_10_ TCID_50_/mL) using TCID_50_ assay and (**D**) viral RNA load (log_10_ RNA copies) using RT-qPCR in limbs and brains. Statistically significant differences after Bonferroni correction are denoted with * *p* < 0.05. (**E**–**I**) H&E staining and IHC analysis of skeletal muscles for (**E**) PBS-inoculated and (**F**) iCVA16-, (**G**) iChi-CCE-, (**H**) iChi-ECE-, and (**I**) iEV-A71-immunised mice followed by CVA16 virus infection. Arrowheads indicate inflammatory cell infiltrates, while arrows indicate viral antigens. Scale bar: 50 µm. Magnification: 10× and 40× (small box).

**Figure 4 vaccines-13-01017-f004:**
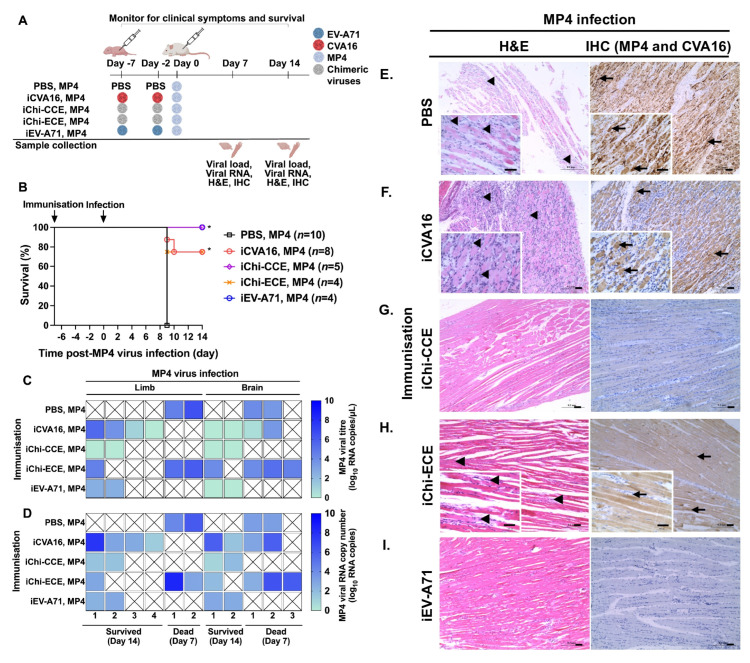
Protective effects of inactivated chimeric viruses against lethal MP4 virus challenge. (**A**) Schematic of *in vivo* experiments to examine the virulence of chimeric viruses. (**B**) Survival percentage of infected mice were recorded. Number of mice are denoted as *n*. Chimeric virus-immunised mice were compared with PBS-inoculated mice. (**C**) Viral titration (log_10_ TCID_50_/mL) using TCID_50_ assay and (**D**) viral RNA load (log_10_ RNA copies) using RT-qPCR in limbs and brains. Statistically significant differences after Bonferroni correction are denoted with * *p* < 0.05. (**E**–**I**) H&E staining and IHC analysis of skeletal muscles for (**E**) PBS-inoculated and (**F**) iCVA16-, (**G**) iChi-CCE-, (**H**) iChi-ECE-, and (**I**) iEV-A71-immunised mice followed by MP4 virus infection. Arrowheads indicate inflammatory cell infiltrates, while arrows indicate viral antigens. Scale bar: 50 µm. Magnification: 10× and 40× (small box).

**Table 1 vaccines-13-01017-t001:** Semiquantitative scoring for pathological lesions and viral antigens in limbs and brains for evaluating viral virulence.

Infection	Staining	Limb		Brain	
		S1	S2	S1	S2
PBS	H&E	–	−	−	−
	IHC	−	−	−	−
CVA16	H&E	++++	++++	−	−
	IHC	++++	+	−	−
Chi-CCE	H&E	+++	+	−	−
	IHC	+++	+	−	−

Pathological lesions in tissues (H&E) were scored as follows: −, no observable lesions; +, slight (1–25%); +++, moderate (51–75%); ++++, severe (>75%) (39). Presence of viral antigen (IHC) in the tissues were graded as follows: −, negative; +, <25% positive; +++, 51–75% positive; ++++, >75% positive; NT, not tested.

**Table 2 vaccines-13-01017-t002:** Semiquantitative scoring for pathological lesions and viral antigens in limbs and brains for evaluating protective efficacy.

Inoculation/Infection		Staining	Limb		Brain	
Day −7	Day 0		Day 7		Day 7	
			S1	S2	S1	S2
PBS	CVA16	H&E	++++	++++	−	−
		IHC	++++	+++	−	−
	MP4	H&E	+++	+++	−	−
		IHC	++++	++++	+	−
iCVA16	CVA16	H&E	−	−	−	−
		IHC	−	−	−	−
	MP4	H&E	++++	+++	−	−
		IHC	++++	−	−	−
iChi-CCE	CVA16	H&E	+	−	−	−
		IHC	−	−	−	−
	MP4	H&E	+	−	−	−
		IHC	−	−	−	−
iChi-ECE	CVA16	H&E	+++	+++	−	−
		IHC	++++	++	−	−
	MP4	H&E	+	+	−	−
		IHC	−	−	−	−
iEV-A71	CVA16	H&E	+	−	−	−
		IHC	−	−	−	−
	MP4	H&E	+	−	−	−
		IHC	−	−	−	−

Pathological lesions in tissues (H&E) were scored as follows: −, no observable lesions; +, slight (1–25%); ++, mild (26–50%); +++, moderate (51–75%); ++++, severe (>75%) [39]. Presence of viral antigen (IHC) in the tissues were graded as follows: −, negative; +, <25% positive; ++, <50% positive; +++, <75% positive; ++++, >75% positive; NT, not tested.

## Data Availability

All data is contained within the article and Appendix A.

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
