# Peer review of "Intertypic Recombination Between Coxsackievirus A16 and Enterovirus A71 Structural and Non-Structural Genes Modulates Virulence and Protection Efficacy"

_vaccines, 2025, doi:10.3390/vaccines13101017_

Round 1

Reviewer 1 Report

Comments and Suggestions for Authors

This study aims to investigate the effects of recombination on replication and pathogenesis of EV-A71. The authors constructed different recombinant viruses comprising various gene segments of CVA16 and EV-A71. Through analyzing the in vitro and in vivo results, Chang et al. demonstrated that CVA16 and recombinant CCE virus were virulent to animals. However, several issues need to be addressed. First, the results showed that infection by CVA16 resulted in death in animals, while EV-A71 was avirulent. This data indicates that newborn BALB/c mice are more susceptible to CVA16 infection and may not be an appropriate animal model to assess the virulence of EV-A71. Second, the results shown in Figure 1 are not essential. It will be much better to remove this part. Third, the number of samples in several experiments is low. For example, in Figures 2C and 2E, only two samples were collected and analyzed in some groups. Fourth, it would be beneficial to quantify the titers of neutralizing antibodies in vaccine studies.

Reviewer 2 Report

Comments and Suggestions for Authors

The manuscript entitled " Intertypic Recombination Between Coxsackievirus A16 and Enterovirus A71 Structural and Non-Structural Genes: Insights into Virulence and Bivalent Vaccine Design” was evaluated. This manuscript presents a well-designed and experimentally robust study investigating the functional consequences of intertypic recombination between CVA16 and EV-A71, two major etiological agents of HFMD. The authors constructed four chimeric viruses by swapping structural (P1) and non-structural (P2/P3) genomic regions between CVA16 and EV-A71 and comprehensively characterized their replication kinetics, virulence in a neonatal mouse model, and potential as bivalent vaccine candidates. The study provides valuable insights into the genetic determinants of virulence and cross-protective immunity, with significant implications for vaccine development. The manuscript is generally well-written, methods are thoroughly described, and the conclusions are supported by the data. Minor revisions are recommended prior to acceptance.

Negative aspects

1.The naming convention for chimeric viruses (e.g., Chi-CCE) is explained but could be summarized in a table for easier reference throughout the text.

2.While statistical methods are appropriately described, some figures (e.g., Figure 2E) show error bars but no significance markers. Ensure all relevant comparisons are clearly indicated.

3.The authors appropriately acknowledge the lack of a 5'UTR-only chimera (Chi-ECC). Elaborate briefly on how this limitation affects the interpretation of results and future directions.

4.The elevation of IL-10 without significant changes in IFN-γ or IL-2 is interesting. Consider discussing whether this suggests a specific immunomodulatory strategy employed by CVA16/Chi-CCE.

5.Minor grammatical errors exist (e.g., “access” should be “assess” on page 14; Line 485). A thorough proofread is recommended.

Reviewer 3 Report

Comments and Suggestions for Authors

Journal: Vaccines (ISSN 2076-393X)

Manuscript ID: vaccines-3865591

Type: Article

Title: Intertypic Recombination Between Coxsackievirus A16 and Enterovirus A71 Structural and Non-Structural Genes: Insights into Virulence and Bivalent Vaccine Design

Authors: Hooi Yee Chang , ... Yoke Fun Chan *

Section: Vaccines against Tropical and other Infectious Diseases

This work focused on the simultaneous infection and genetic recombination of the two main types of viruses causing hand, foot, and mouth disease (HFMD), created chimeras of them, and evaluated their pathogenicity and effectiveness as vaccines.

This reviewer finds that the experimental results presented in this work are clear, and the research on changes in viral toxicity due to recombination and mutation is interesting. However, there were parts that required additional explanation. Therefore, this reviewer would like to recommend publication in 'Vaccines' after the necessary revisions have been sufficiently made.

Comments

While the experiments, their results, and the discussion are very clear, this reviewer felt there were items where the explanations following parts mentioned in the introduction were insufficient. Please address these sufficiently to reduce reader confusion.

  1. Fig.1A shows the structure of the chimeras created in this study.

L52-54 introduces the functions of the gene products produced by the ORFs. Please describe the "structural protein" and "non-structural protein" in Fig.1A more prominently. Since the functions of these gene products are relevant for the interpretation of the authors' experimental results, mapping the simple ORF products to the three regions used for chimera creation would be beneficial for readers. One focus of this study, as stated in L414-415, is "to examine the impact of structural and non-structural protein exchanges". This reviewer understands that descriptions of individual gene products are around L100-105.

  1. L63-66 states that EVs exist in types A-D, and the text mentions that A71 is used in the study. An explanation is needed regarding the potential differences between Group I and Group II, and their relevance to this paper. In particular, the authors emphasize the importance of the 5'UTR in the discussion from L465 onwards.

  1. In L71-72, the authors state, "These recombination events are not random but occur at these hotspots which are more conserved across different enteroviruses." Please add explanatory text regarding the relevance to the creation of the Chis in this study.

  1. In L484-486, the authors state, "We did not include a construct with 5’UTR substitution alone (Chi-ECC) which limits our ability to access the independent contribution of 5’UTR to the observed phenotypic changes." Please also state the reason for not including constructs like Chi-CEE this time.

  1. Regarding the protective effect of iChi-CEE (3.4), the text evaluates histochemistry, survival, and viral load. Since this journal is about vaccines, if there is data on antibody production or actual cytokines, please mention it.

Round 2

Reviewer 3 Report

Comments and Suggestions for Authors

This work focused on the simultaneous infection and genetic recombination of the two main types of viruses causing hand, foot, and mouth disease (HFMD), created chimeras of them, and evaluated their pathogenicity and effectiveness as vaccines.This reviewer finds that the experimental results presented in this work are clear, and the research on changes in viral toxicity due to recombination and mutation is interesting. It is unfortunate that there is no data regarding the acidity of the antibody, the authors have adequately addressed all comments and made the necessary revisions. Therefore, this reviewer recommends the acceptance of the paper in “Vaccines”.